

# Benchmarking of a time-saving and scalable protocol for the extraction of DNA from diverse viromes

Michael Shamash[1,2], Saniya Kapoor[1] and Corinne F. Maurice[1,2]

[1] Department of Microbiology & Immunology, McGill University, Montreal, QC, Canada
[2] McGill Centre for Microbiome Research, Montreal, QC, Canada

## ABSTRACT

The virome, composed of viruses inhabiting diverse ecosystems, significantly influences microbial community dynamics and host health. The phenol-chloroform DNA extraction protocol for viromes, though effective, is time-intensive and requires the use of multiple toxic chemicals. This study introduces a streamlined, scalable protocol for DNA extraction using a commercially-available kit as an alternative, assessing its performance against the phenol-chloroform method across human fecal, mouse fecal, and soil samples. No significant differences in virome diversity or community composition were seen between methods. Most viral operational taxonomic units (vOTUs) were common to both methods, with only a small percentage unique to either approach. Alpha- and beta-diversity analyses showed no significant impact of the extraction method on virome composition, confirming the kit's efficacy and versatility on sample types beyond those officially supported by the manufacturer. While the kit approach offers benefits like reduced toxicity and increased throughput, it has limitations such as higher costs and potential issues reliably capturing low-abundance taxa. This protocol provides a viable option for large-scale virome studies, although the phenol-chloroform approach may still be preferable for specific sample types.

## INTRODUCTION

Viromes, the collections of viruses inhabiting diverse ecosystems, play important roles in shaping microbial community composition and function (*Reyes et al., 2013*; *Breitbart et al., 2018*; *Hsu et al., 2019*; *Howard-Varona et al., 2020*; *Graham et al., 2024*). In host-associated ecosystems, like the human microbiome, the virome is also a key determinant of host health (*Liang & Bushman, 2021*).

Despite significant advances in the field in recent years, there remain several challenges associated with studying viromes. Many of these challenges are computational, such as identifying novel and highly divergent viruses, or characterizing the host range of uncultured viruses (*Khan Mirzaei et al., 2021*). Others are related to sample processing, especially in low biomass sample types, such as amplifying low amounts of purified virome nucleic acids or dealing with high levels of background contamination (usually of human, mouse, or bacterial origin) (*Khan Mirzaei et al., 2021*).

Corresponding author
Corinne F. Maurice,
corinne.maurice@mcgill.ca

It is important that virome nucleic acid extraction protocols be efficient and reliable so that we may begin to address some of these analytical challenges. The phenol-chloroform DNA extraction protocol is currently the gold standard for studying viral metagenomes (*Thurber et al., 2009*; *Shkoporov et al., 2018*; *Göller et al., 2020*; *Sinha et al., 2022*). However, this protocol has its limitations, including being time-consuming and requiring the use of multiple toxic organic chemicals such as phenol and chloroform.

Given the growing interest in studying viromes across diverse environments, including in human (*Shkoporov & Hill, 2019*; *Shamash & Maurice, 2021*), animal (*Brunse et al., 2021*; *Shkoporov et al., 2022*), and soil (*Roux & Emerson, 2022*) communities, there is a need for a faster and scalable extraction protocol which can accommodate increasingly large studies.

In this study, we set out to benchmark a streamlined protocol for virome DNA extractions using a commercially available column-based DNA extraction kit. While the manufacturer recommends its use for plasma, serum, and cell-free body fluids, we apply the kit to extract viral DNA from such as stool and soil samples. Although this kit has been used before for extracting viral DNA from human fecal samples, its performance has not been compared to that of the phenol-chloroform protocol (*Bikel et al., 2022*). Here we compare protocols, emphasizing the time savings and avoidance of toxic organic reagents when using the kit. Overall, we find no significant differences in the resulting sequenced viromes between the two methods, with comparable sample yield, viral diversity, and community composition.

## METHODS

Portions of this text were previously published as part of a preprint (*Shamash, Kapoor & Maurice, 2024*).

### Sample collection

Human fecal samples were collected with the approval of protocol A04-M27-15B from the McGill University Institutional Review Board. Participants provided informed written consent for the utilization of their samples and met specific inclusion criteria: they were 18 years of age or older, had no diagnosed gastrointestinal disease, and had not used antibiotics in the 3 months prior to sampling. Fresh fecal samples were collected, aliquoted in an anaerobic chamber, and kept at −70 °C until processing.

Mouse fecal samples were collected with the approval of animal use protocol MCGL-7999 from McGill University. Fresh fecal samples were collected and kept at −70 °C until processing. All mice had unlimited access to standard chow and water.

Soil samples were collected in sterile 50 mL conical centrifuge tubes from various locations on the downtown campus of McGill University (coordinates: 45.5042 N, 73.5755 W).

### Sample pre-processing

Prior to extracting virome DNA, we first enriched for virus-like particles (VLPs) in the fecal and soil samples. Samples were resuspended in sterile (0.02 μm-filtered) PBS as follows: human fecal samples, ~200–400 mg in 2 mL PBS; mouse fecal samples, ~100 mg in 1 mL PBS; soil samples, 10 mL soil in 10 mL PBS (1:1 volume ratio). Large debris was

pelleted by centrifugation at 1,000 g for 5 min, and the supernatant recovered. Bacterial cells were pelleted by centrifugation at 10,000 g for 10 min. A total of 500 μL of the VLP-containing supernatant was added to a sterile Ultrafree-MC centrifugal filter unit (MilliporeSigma, Burlington, MA, USA) with 0.22 μm pore size and centrifuged at 12,000 g for 2 min, and 400 μL of purified VLPs was recovered. The VLP suspension was cleaned by addition of 100 μL chloroform (final chloroform concentration: 20% v/v), followed by thorough vortexing and centrifugation at 21,000 g for 5 min. The VLP-containing supernatant (upper layer) was carefully recovered and transferred to a new tube containing 5 μL (10 U) TURBO DNase (Thermo Fisher Scientific, Waltham, MA, USA), 50 μL 10X TURBO DNase buffer, and 1 μL (approx. 125 U) Benzonase DNase (MilliporeSigma, Burlington, MA, USA). The sample was incubated at 37 °C with mild shaking for 90 min. To stop the DNase digestion reaction, 18 μL of a 500 mM EDTA (pH 8) solution was added, followed by heat inactivation of the enzymes at 75 °C for 30 min. Sterile PBS was added to bring the samples up to 800 μL in volume: 400 μL for phenol/chloroform extraction, and 400 μL for kit extraction.

## DNA extraction—phenol/chloroform approach

This protocol is adapted from *Thurber et al. (2009)*, where 40 uL of 200X TE buffer (2M tris, 200 mM EDTA, pH 8.5), 440 uL formamide, and 10 uL UltraPure glycogen (20 mg/mL stock; Thermo Fisher Scientific, Waltham, MA, USA) were added to 400 μL purified VLPs and incubated at room temperature for 30 min. Two volume equivalents (approx. 1,780 μL) of room temperature 100% ethanol were added to the sample, and DNA was pelleted by centrifugation at 10,000 g for 20 min at 4 °C. The supernatant was carefully removed and discarded, and the crude DNA pellet was washed twice with 1 mL of ice-cold 70% ethanol, centrifuging as above between washes. The final pellet was dried for 5 min at room temperature and resuspended in 567 μL of 1X TE buffer (10 mM tris, 1mM EDTA, pH 8). We then added 30 μL of 10% (w/v) SDS and 3 μL of Proteinase K (20 mg/mL stock; Thermo Fisher Scientific, Waltham, MA, USA) to the sample and briefly vortexed before incubation at 45 °C for 1 h with gentle shaking. After incubation, 100 μL of 5M NaCl and 80 μL of CTAB buffer (1.1 M NaCl, 450 mM CTAB/cetyltrimethylammonium bromide) were added, the sample vortexed and incubated at 65 °C for 10 min with gentle shaking. One volume equivalent (approx. 780 μL) of chloroform:isoamyl alcohol 24:1 (Sigma-Aldrich) was added and mixed well, transferred to a light phase lock gel tube (PLG tube; Quantabio, Beverly, MA, USA), and centrifuged at 12,000 g for 5 min. The aqueous (upper) phase was transferred to a new PLG tube to which another 1 volume equivalent of phenol:chloroform: isoamyl alcohol 25:24:1 (pH 8, Invitrogen) was added, mixed well, and centrifuged as above. After transferring the aqueous (upper) phase to a new PLG tube, this step was repeated with 1 volume equivalent of chloroform:isoamyl alcohol 24:1. The aqueous phase was then recovered, added to a tube containing 550 μL (~0.7 volume equivalents) ice-cold 100% isopropanol, and stored overnight at −20 °C for DNA precipitation. The next day, DNA was pelleted by centrifugation at 13,000 g for 15 min at 4 °C. The pellet was washed once with 500 μL ice-cold 70% ethanol and pelleted again as above. The final pellet was air-dried at

room temperature, resuspended in 50 μL tris buffer (10 mM Tris-Cl, pH 8), and stored in a DNA LoBind tube (Eppendorf, Hamburg, Germany) at −20 °C until library preparation.

## DNA extraction—kit approach

DNA was extracted using the QIAGEN MinElute Virus Spin Kit (QIAGEN, Hilden, Germany) according to the manufacturer's instructions, with the following modifications to incorporate a larger sample input volume. Fifty μL QIAGEN Protease and 400 μL Buffer AL were added to 400 μL purified VLPs, vortexed, and incubated at 56 °C for 15 min. We then added 500 μL 100% ethanol to the sample, vortexed, and incubated at room temperature for 5 min. Half of the sample (~675 μL) was added onto a QIAamp MinElute column, and centrifuged at 6,000 g for 1 min. The filtrate was discarded and the remaining sample was added to the same QIAamp MinElute column, centrifuged as above. The remainder of the protocol remained unchanged from the manufacturer's instructions. DNA was eluted from the column using 50 uL Buffer EB (tris buffer; 10 mM Tris-Cl, pH 8; 5-min incubation before elution), and stored in a DNA LoBind tube (Eppendorf, Hamburg, Germany) at −20 °C until library preparation. The included Carrier RNA was not used for any of the DNA extractions.

## Library preparation and sequencing

Extracted vDNA (5 μL per sample) from both procedures was quantified using the Qubit 1X dsDNA High Sensitivity kit (Invitrogen). The Qubit instrument was re-calibrated before each use using the included standards. Sequencing libraries were prepared using the Illumina DNA Prep kit (Illumina, San Diego, CA, USA) according to the manufacturer's instructions, maximizing sample input (30 μL) and number of PCR cycles (12 cycles). The final libraries were quantified using the Qubit 1X dsDNA High Sensitivity kit and the Bioanalyzer High Sensitivity DNA kit (Agilent Technologies, Santa Clara, CA, USA). An equimolar pool of libraries was created and sequenced on an Illumina MiSeq instrument with 150 bp paired-end reads (SeqCenter, Pittsburgh, PA, USA).

## Bioinformatic analysis of sequence data

Raw reads were trimmed and filtered with fastp (v0.20.1) (*Chen et al., 2018*) using the following criteria: –detect_adapter_for_pe -q 15 –cut_right –cut_window_size 4 –cut_mean_quality 20 –length_required 100. Trimmed reads were decontaminated for human and mouse genomic DNA with bowtie2 (v2.4.2) (*Langmead & Salzberg, 2012*) using the *Homo sapiens* GRCh38 and *Mus musculus* GRCm39 references, respectively. metaSPAdes (v3.15.4) (*Nurk et al., 2017*) was used to conduct *de novo* assembly of each sample individually using default settings. VIBRANT (v1.2.1) (*Kieft, Zhou & Anantharaman, 2020*) was used on these filtered contigs to identify viral sequences with the following settings: 3 kb minimum length, virome mode. Viral contigs were dereplicated into viral operational taxonomic units (vOTUs) using BLASTN (v2.14.0) (*Camacho et al., 2009*) followed by the anicalc.py and aniclust.py scripts from CheckV (*Nayfach et al., 2021*) with the following parameters: 95% average nucleotide identity (ANI) over 85% of the shorter contig's length. Bowtie2 (v2.4.2) (*Langmead & Salzberg, 2012*) was used to map

decontaminated trimmed/filtered reads to the resulting dereplicated set of vOTUs, and a coverage summary report was generated with Samtools (v1.13) (*Li et al., 2009*) using the 'samtools coverage' command. A standard vOTU detection threshold was applied prior to subsequent our analyses: mean depth of coverage ≥1X and breadth of coverage ≥75% (*Roux et al., 2017*). iPHoP (v1.3.3) (*Roux et al., 2023*) was used to computationally predict hosts of vOTUs. R (v4.2.2) was used for all diversity analyses with the following packages: phyloseq (v1.42.0) (*McMurdie & Holmes, 2013*) and vegan (v2.6.4) (*Oksanen et al., 2019*). Flextable (v0.8.5) (*Gohel & Skintzos, 2024*) was used to generate tables.

## RESULTS

To compare the kit-based viral DNA extraction protocol (KIT) with the previous standard, phenol-chloroform extractions (PC), we collected samples from a variety of environments which were each processed with both extraction methods: two soil samples, seven human fecal samples, and four mouse fecal samples (Fig. 1). DNA yields were not significantly different across protocols for soil (42 ± 2 pg/μL KIT, 47 ± 32 pg/μL PC), human fecal (249 ± 141 pg/μL KIT, 536 ± 314 pg/μL PC) and mouse fecal (45 ± 14 pg/μL KIT, 126 ± 42 pg/μL PC) samples ($p > 0.05$, Wilcoxon signed-rank test).

After sequencing, assembly, and viral detection, a total of 963, 707, and 423 vOTUs were detected in human fecal, mouse fecal, and soil samples, respectively. Of these, most vOTUs were common to both PC and KIT viromes: 875 (91%) of all vOTUs in human fecal viromes, 630 (89%) of all vOTUs in mouse fecal viromes, and 398 (94%) of all vOTUs in soil viromes (Fig. 2A). Few vOTUs (9% of human fecal, 11% of mouse fecal, and 6% of soil vOTUs) were detected exclusively in either PC or KIT viromes (Fig. 2A). Assembly statistics (*e.g.*, total assembly size and N50) and vOTU read recruitment percentages were consistently similar between KIT and PC approaches (Table S1).

We next characterized the alpha- and beta-diversity of our samples, to evaluate compositional differences in viromes which may be due to extraction method. Alpha diversity, measured at the observed richness and Shannon diversity levels, was consistently similar between both extraction methods (mean 255, 317, 200 observed vOTUs, and 3.86, 4.05, 5.10 Shannon index for human fecal, mouse fecal, and soil viromes, respectively, with KIT; and mean 233, 384, 210 observed vOTUs, and 3.82, 4.14, 5.28 Shannon index for human fecal, mouse fecal, and soil viromes, respectively, with PC), with these differences being insignificant (Fig. 2B). Pairwise Bray-Curtis distances were calculated between all samples and plotted on an NMDS plot (Fig. 2C). For each sample, PC and KIT viromes clustered closely together regardless of the environment. A PERMANOVA analysis confirmed that the distances between samples was explained primarily by environment ($R^2 = 0.363$, $p = 0.001$) and the sample itself ($R^2 = 0.613$, $p = 0.001$), rather than the extraction method ($R^2 = 0.002$, $p = 0.426$; Fig. 2C).

Finally, we predicted the bacterial hosts of vOTUs to verify that the predicted hosts align with bacteria associated with the sampled environments. The most abundant soil virome vOTUs with assigned hosts were predicted to infect members of the *Mycobacteriales*, *Flavobacteriales*, and *Streptomycetales*, *Pseudomonadales* orders. In human fecal samples,

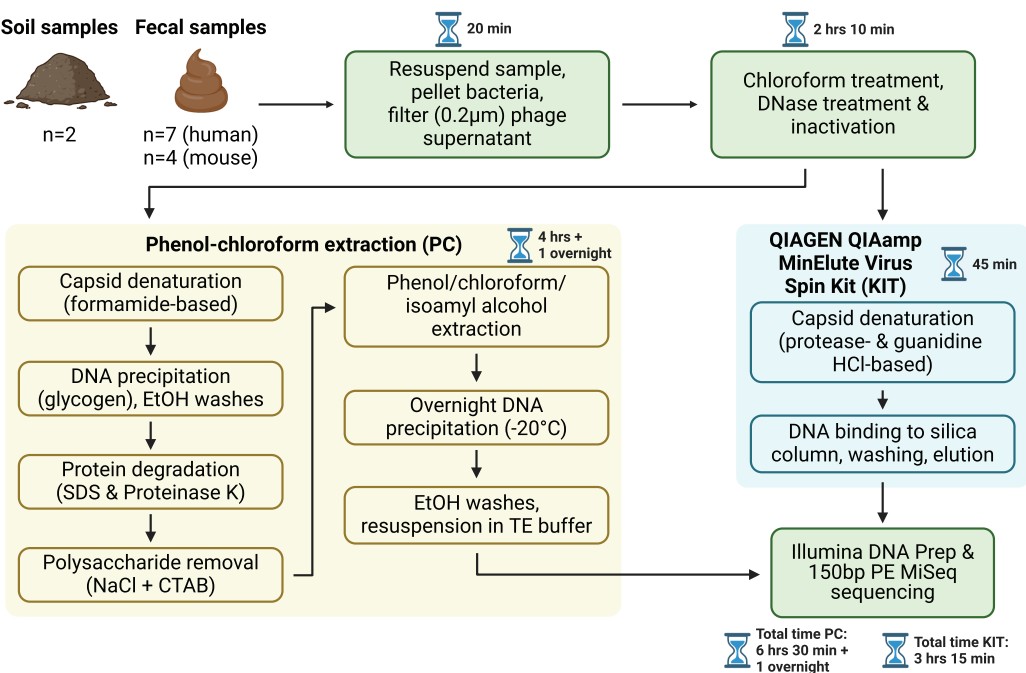

**Figure 1 Overview of sample processing pipeline.** All samples underwent resuspension in sterile PBS, filtration to remove debris and bacterial or eukaryotic cells, chloroform treatment, and DNase treatment (with subsequent DNase inactivation). Samples were then split to be processed with either the phenol-chloroform extraction (PC) or QIAGEN QIAamp MinElute Virus Spin Kit (KIT) DNA extraction protocols. DNA libraries were prepared with the Illumina DNA Prep kit and sequenced on an Illumina MiSeq instrument with 150 bp paired-end reads. Approximate sample processing times are indicated for each step. The KIT protocol increases sample throughput by reducing protocol complexity and minimizing the number of incubation steps. Created in BioRender (*Maurice, 2024*).

the most abundant vOTUs with assigned hosts infect were predicted to infect members of the *Bacteroidales*, *Lactobacillales*, *Oscillospirales*, and *Lachnospirales* orders. In mouse fecal samples, the most abundant vOTUs with assigned hosts were predicted to infect members of the *Lachnospirales*, *Oscillospirales*, *Bacteroidales*, and *Peptostreptococcales* orders.

## DISCUSSION

In this study, we describe a streamlined protocol for virome DNA extractions using a commercially available column-based DNA extraction kit, with some additional upstream steps. Unlike the manufacturer's recommended use for cell-free body fluids, plasma, or serum, we applied this protocol to more complex sample types, including human fecal, mouse fecal, and soil samples. This approach offers several advantages over the gold standard phenol-chloroform protocol, including reduced dependency on toxic organic chemicals, increased throughput, and improved ease-of-use. We compared this kit's performance to that of the phenol-chloroform protocol to evaluate its efficacy across these diverse sample types (Fig. 1). Our results show that the choice of protocol did not significantly affect virome diversity or community composition (Fig. 2).

Most of the assembled vOTUs were detected in both KIT and PC viromes, with few vOTUs unique to either approach (Fig. 2A). The presence of unique vOTUs to either KIT

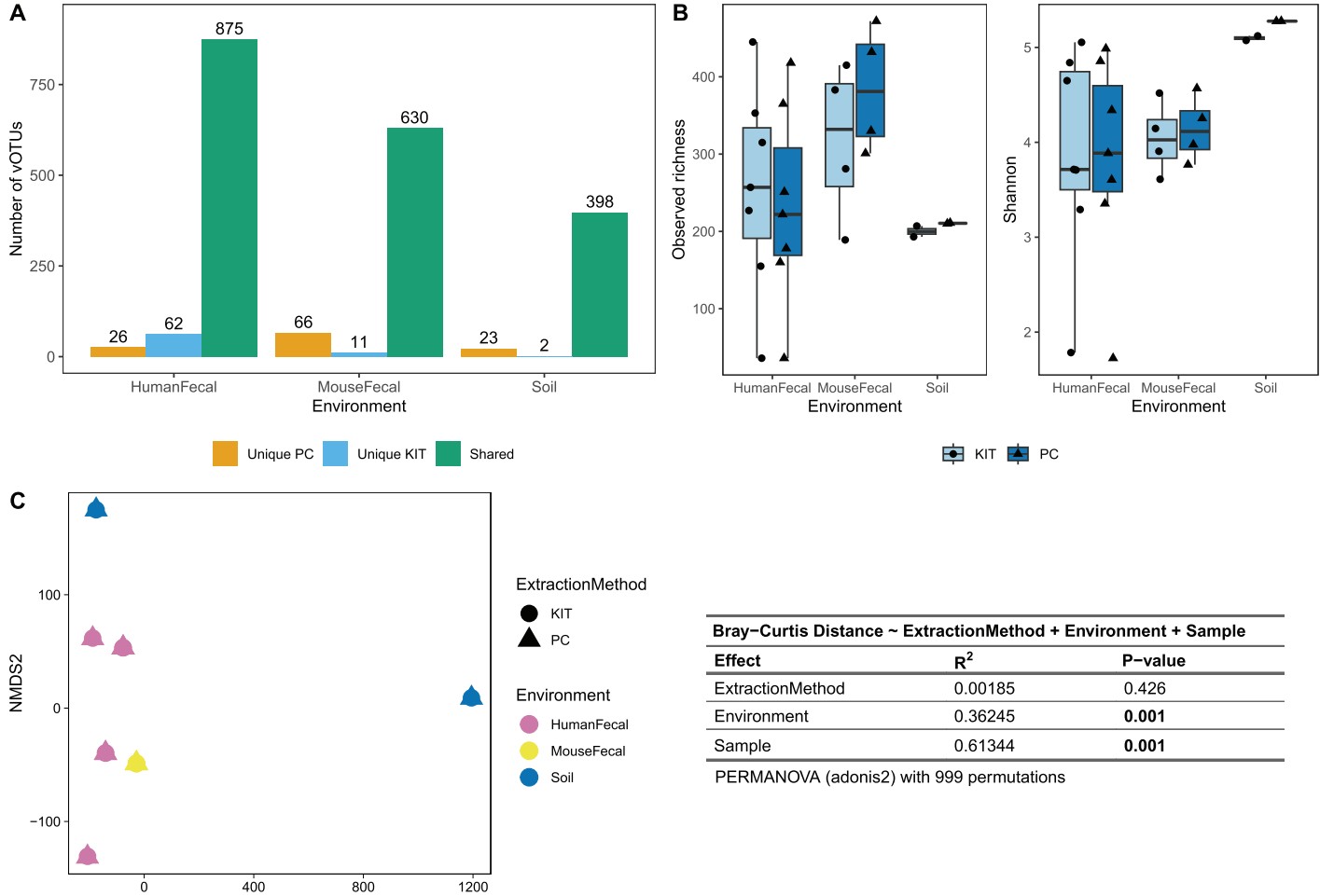

**Figure 2 PC and KIT extraction methods result in equivalent virome communities.** (A) The number of vOTUs detected only within the phenol-chloroform (PC), or kit (KIT) extraction method, or shared between both methods. (B) Observed richness (left) and Shannon diversity (right) of viromes according to environment and extraction method. No statistical differences were observed between extraction methods within any environment (Wilcoxon signed-rank test, $p > 0.05$). (C) Non-metric multidimensional scaling (NMDS) of Bray-Curtis distances of vOTUs between samples, colors representing sample environment and shapes representing extraction method (NMDS stress = $9.4 \times 10^{-5}$). Tests for significant differences in Bray-Curtis dissimilarity were conducted using PERMANOVA (adonis2) with 999 permutations, and summary statistics are reported in the table.

or PC viromes did not have a significant effect on community alpha- or beta- diversity (Figs. 2B, 2C). The unique vOTUs were low in abundance (mean unique vOTU abundance of 0.13 ± 0.42%) and made up a small fraction of the overall viromes (mean cumulative unique vOTU abundance of 1.75 ± 0.26%), indicating that both extraction methods were equally good at capturing high-abundance vOTUs. vOTUs host prediction was consistent with descriptions of soil (*Graham et al., 2024*), adult human gut (*Camarillo-Guerrero et al., 2021*), and adult mouse gut (*Ritz et al., 2024*) viromes.

While this kit has been used to extract viral DNA from human fecal samples in other published work, there has been no direct comparison against the phenol-chloroform gold standard, or any other extraction protocol (*Bikel et al., 2022*). In addition, there are several different experimental steps between both studies. Specifically, we suggest using two

DNases to maximize removal of bacterial and host DNA, inactivating the nucleases with EDTA and a higher incubation temperature (75 °C for 30 min) before proceeding directly to extraction with the kit, and allowing the column-bound DNA to sit with the elution buffer for 5 min. All these steps aim to maximize DNA yield and remove non-viral DNA.

Despite its advantages, the kit approach has a few limitations. This kit still includes a chloroform step upstream of extraction which destroys the membrane of enveloped viruses (*Callanan et al., 2021*). Cost may be another factor, as kits are generally more expensive than the different reagents used in the phenol-chloroform method. Furthermore, reliance on proprietary reagents may be an issue if the manufacturer changes their formulation. Finally, while kits offer high consistency, they may not always result in the highest yield for all sample types, as we report here for soil samples. Researchers specifically targeting low-abundance virome members may still wish to use the phenol-chloroform method. For researchers working with low biomass samples, it is important to test and validate the kit's performance with this sample type and include reagent controls.

## ACKNOWLEDGEMENTS

ChatGPT (OpenAI Inc., San Francisco, CA, USA) was used for editing purposes to improve clarity of a few sentences. The sentence meaning was not altered, and there were no changes to any of our presented data, facts, or conclusions by using this tool.

### Funding

This work was funded by a Canadian Institutes of Health Research Canada Project grant (PJT-175065) and an NSERC Discovery grant (RGPIN 2023-04216) to Corinne F. Maurice. Corinne F. Maurice is a Tier 2 Canada Research Chair in gut microbial interactions. Michael Shamash is supported by the Canadian Institutes of Health Research Canada Graduate Scholarship to Honor Nelson Mandela (CIHR CGS-D; #DF2-187718), and the Fonds de recherche du Québec-Santé: Bourse de formation au doctorat (FRQS; #311071). The funders had no role in study design, data collection and analysis, decision to publish, or preparation of the manuscript.

### Grant Disclosures

The following grant information was disclosed by the authors:
Canadian Institutes of Health Research Canada: PJT-175065, CIHR CGS-D; #DF2-187718.
NSERC Discovery: RGPIN 2023-04216.
Fonds de recherche du Québec-Santé: Bourse de formation au doctorat: FRQS; #311071.

### Competing Interests

Corinne F. Maurice is an Academic Editor for PeerJ.

## Author Contributions

- Michael Shamash conceived and designed the experiments, performed the experiments, analyzed the data, prepared figures and/or tables, authored or reviewed drafts of the article, and approved the final draft.
- Saniya Kapoor conceived and designed the experiments, performed the experiments, prepared figures and/or tables, and approved the final draft.
- Corinne F. Maurice conceived and designed the experiments, analyzed the data, prepared figures and/or tables, authored or reviewed drafts of the article, and approved the final draft.

## Human Ethics

The following information was supplied relating to ethical approvals (*i.e.*, approving body and any reference numbers):

Research Ethics Office (IRB) of the Faculty of Medicine and Health Sciences at McGill University.

## Animal Ethics

The following information was supplied relating to ethical approvals (*i.e.*, approving body and any reference numbers):

Comparative Medicine and Animal Resources Centre Facility Animal Care Committee (CMARC FACC) at McGill University.

## Data Availability

The virome sequencing reads (human and mouse sequences removed) are available at NCBI SRA: PRJNA1125394.

The code and scripts are available at Zenodo: Michael Shamash. (2024). mshamash/vdna_protocol_manuscript: Manuscript_v2 (Version v2). Zenodo. https://doi.org/10.5281/zenodo.14236083.

## Supplemental Information

Supplemental information for this article can be found online at http://dx.doi.org/10.7717/peerj.18785#supplemental-information.

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
