# Peer review of "Benchmarking of a time-saving and scalable protocol for the extraction of DNA from diverse viromes"

_PeerJ, doi:10.7717/peerj.18785_

## Round 0.1 · original submission · Major Revisions

Reviewer 1 found no notable differences between this work and a previously published study of the authors. Both in the rebuttal and in the text this critics must be discussed in details. I believe that the authors will be able to address all other comments of all reviewers as well.

Reviewer 1 ·

Basic reporting

I have thoroughly reviewed the current work and found no notable differences between it and a previously published study that employs a very similar experimental method and kit to investigate the human virome (https://pubmed.ncbi.nlm.nih.gov/35199035/). The research objectives and methodologies in both studies closely align, raising essential questions about the novelty and contribution of the current findings.

I recommend that the authors introduce and actively discuss their work compared to the previously published study. This discussion should include a detailed comparison of the methodologies used, highlighting similarities and identifying areas where they have built upon or diverged from the earlier research. Such a comparative analysis will help to contextualize their findings within the broader landscape of virome research, enhancing the significance of the current study and providing a clearer understanding of its contributions to the field. Addressing these points will strengthen the manuscript and offer readers a more comprehensive view of the subject matter.

Experimental design

no comment

Validity of the findings

no comment

Additional comments

no comment

·

Basic reporting

Very well written and perfectly understandable.

Experimental design

No comment

Validity of the findings

No comment

Additional comments

Line 31: Tenses don't read very well, maybe "Viromes, the collections of viruses inhabiting diverse ecosystems.."
Line 74: Explain VLPs on first use
Line 100: I found it a little confusing that this was termed the 'DNA pellet' at this early stage
Line 171: Consider 'the' instead of 'our'

Reviewer 3 ·

Basic reporting

In this manuscript, Shamash et al. compare the efficacy of two viral DNA extraction methods: a phenol/chloroform-based protocol, which has been considered the "gold standard," and a commercially available kit developed by Qiagen.

The manuscript is well written, and the conclusions are straightforward. I appreciate the detailed methods section, which makes the experiments appear easily replicable. Additionally, I was able to easily access the raw data and code. The information presented in the paper appears useful as extracting viromes can be a lengthy and difficult process. However, before publication, I have a few concerns I believe the authors should address, along with some suggestions for improvement.

Specific Points:

I believe the title and parts of the text are misleading (e.g. line 171, "To compare our new viral DNA protocol"). At first glance, I expected a completely new extraction method rather than the comparison of a manufactured kit for viral extraction. The title should clarify that this study is a benchmarking of two methods, including an initial viral enrichment step, rather than a new development from scratch.

Additionally, the authors could emphasize that they are applying the kit to extract viruses from sources not originally specified in the kit description (e.g., plasma, serum, and cell-free body fluids), along with noting the time savings and avoidance of organic reagents, which are already highlighted as benefits on the Qiagen website (https://www.qiagen.com/us/products/discovery-and-translational-research/dna-rna-purification/multianalyte-and-virus/qiaamp-minelute-virus-kits). This would help clarify the knowledge gap that is being filled and purpose of the paper.

The results (lines 179-184) discussed unique and shared vOTUs in the samples, and then also uses the word "contig" The associate figure (2A) is labeled "viral contigs". I would interpret the viral contig as that which is identified by VIBRA NT and then the vOTUs are identified after the BLASTN and CheckV clustering. It seems this section is referring solely to the vOTUs? The language here is a bit confusing and could be clarified .

Experimental design

1) The introduction (lines 38–39) mentions that contamination can be an issue with viral DNA extraction and analysis. Was potential level of contamination evaluated for each of these methods? Additionally, the yields for some samples were quite low, as is often the case with viromes. Were any negative controls (like a reagent only control) used to check for background DNA? This could help determine whether the detected viruses genuinely represent their source environment. While I am convinced that these methods perform equally well, I wonder if either of them is particularly accurate, especially for low biomass environments like soil.

Another suggestion would be to use a host prediction program, such as iPhop, to assess whether the predicted viruses are likely to infect hosts associated with the sampled environment. At the very least, I think this limitation should be added to the discussion.

2) Was any quality assessment or filtering performed on the viral contigs? A 1 kb cutoff is relatively low low. If the data is enriched in low-quality, small viral contigs, this could affect the accuracy of vOTU clustering and diversity analyses. If the 1kb cut off is used, some kind of quality analysis should be included.

3) Adding assembly statistics would strengthen the manuscript. While these might be inferred from the raw data, presenting them explicitly would clarify that the DNA and assembly results are comparable across methods both in yield and quality of that yield .

Validity of the findings

No comment, covered in previous sections.

Additional comments

Smaller, additional comments:

1) Including a figure that directly compares the (hands-on) time required for each protocol could enhance the appeal of the Qiagen kit method. Does the kit save hours or just minutes?

2) It would be helpful to specify the sample amount used in the Qubit assay. A reading of 42 pg is technically below the Qubit 1x HS detection range and could only be obtained if a larger sample volume was used and then the concentration back calculated. Adding in volume of sample used to Qubit would help confirm that the 0.042 ng/µl reading is not an artifact from Qubit reagents or a mis-calibration of equipment.

4) In Figure 2B, the Y-axis label could be more specific. For example, replacing "observed" with "observed richness" would improve clarity.

---

## Round 0.2 · accepted · Accept

I am happy that the authors have addressed all of the reviewers' comments. I assume that this manuscript is ready for publication.

Reviewer 1 ·

Basic reporting

The authors have fixed all the issues.

Experimental design

The authors have fixed all the issues.

Validity of the findings

The authors have fixed all the issues.

Reviewer 3 ·

Basic reporting

I am happy with the changes made by the authors and believe the manuscript to be improved. I have only a couple of minor comments:

1) It would be good to include the raw iPhop data somewhere.
2) Biorender recently updated their citation guidelines, it may be worth double checking: https://help.biorender.com/hc/en-gb/articles/17605511350685-How-to-cite-your-BioRender-figure

Experimental design

N/A

Validity of the findings

N/A

Additional comments

N/A